# Validation in an Independent Cohort of MiR-122, MiR-1271, and MiR-15b as Urinary Biomarkers for the Potential Early Diagnosis of Clear Cell Renal Cell Carcinoma

**DOI:** 10.3390/cancers14051112

**Published:** 2022-02-22

**Authors:** Giovanni Cochetti, Luigi Cari, Vincenza Maulà, Rosy Cagnani, Alessio Paladini, Michele Del Zingaro, Giuseppe Nocentini, Ettore Mearini

**Affiliations:** 1Division of Urology Clinic, Department of Medicine and Surgery, University of Perugia, 06129 Perugia, Italy; giovanni.cochetti@unipg.it (G.C.); vincenza.maula@unipg.it (V.M.); rosy.cagnani@unipg.it (R.C.); alessio.paladini@ospedale.perugia.it (A.P.); michele.delzingaro@unipg.it (M.D.Z.); ettore.mearini@unipg.it (E.M.); 2Pharmacology Section, Department of Medicine and Surgery, University of Perugia, 06129 Perugia, Italy; luigi.cari@unipg.it

**Keywords:** clear cell renal cell carcinoma (ccRCC), diagnosis, urinary marker, microRNA (miRNA), quantitative reverse transcription PCR (qRT-PCR)

## Abstract

**Simple Summary:**

The survival of patients with the most common type of kidney cancer (called Clear cell renal cell carcinoma—ccRCC) would dramatically improve if it was diagnosed earlier. Early diagnosis can be achieved using imaging techniques, but they are too expensive and therefore cannot be used to screen the population at risk for ccRCC. A few months ago, we published a study that evaluated the amount of certain small RNAs present in urine and showed that they are present at different levels in the urine of ccRCC patients vs. healthy subjects, and based on this discrepancy, we developed an algorithm that can anticipate the presence of kidney cancer. Such studies, however, can suffer from a technical bias called overfitting, such that the method may seem predictive even when it is not. In the present study, we sought to address this possibility and evaluate the amount of the same small RNAs in the urine of an independent cohort. As a result, we demonstrate that the previously developed algorithm has a sensitivity of 96% and specificity of 65%, thus validating this technique for potential application in the early diagnosis of ccRCC with a noninvasive assay.

**Abstract:**

Clear cell renal cell carcinoma (ccRCC) is the most common type of renal cell carcinoma, and the absence of symptoms in the early stages makes metastasis more likely and reduces survival. To aid in the early diagnosis of ccRCC, we recently developed a method based on urinary miR-122-5p, miR-1271-5p, and miR-15b-5p levels and three controls. The study here presented aimed to validate the previously published method through its application on an independent cohort. The expression of miRNAs in urine specimens from 28 ccRCC patients and 28 healthy subjects (HSs) of the same sex and age was evaluated by RT-qPCR. Statistical analyses were performed, including the preparation of receiver operating characteristic (ROC) curves. The mean ccRCC diameter in ccRCC patients was 4.2 ± 2.4 mm. Urinary miRNA levels were higher in patients than in HSs. The data were processed using the previously developed algorithm (7p-urinary score), and the area under the curve (AUC) of the algorithm’s ROC curve was 0.81 (*p*-value = 0.0003), with a sensitivity of 96% and specificity of 65%. Therefore, the 7p-urinary score is a potential tool for the early diagnosis of ccRCC.

## 1. Introduction

Renal cell carcinoma represents about 3–5% of all tumors, and Western countries account for 50% of the global incidence. In the last two decades, there has been a 2% growth in the annual incidence of RCC both worldwide and in Europe [1,2,3]. In 2020, over 430,000 cases of RCC were diagnosed in the world and about 180,000 people died from this disease [4]. Moreover, 138,611 new cases were detected in Europe and 54,054 cancer-related mortalities were recorded [2]. About 70% of renal cell carcinoma showed clear cell renal cell carcinoma (ccRCC) histology.

The clinical peculiarity of renal cell carcinoma (RCC) is the lack of symptoms at earlier stages which allows a silent growth of the tumor and thus about 20% of RCC show metastasis at the time of diagnosis leading to a dramatic reduction in patient survival [5]. For this clinical feature, over 60% of RCC are detected incidentally despite the increasing diffusion of technologically advanced imaging methods. The RCC prognosis has been proved to be associated with tumor staging: when the tumor size is below 3 cm the risk of metastasis is lower than 5%, 18% when the diameter is of 6–7 cm, whereas in advanced-stage disease the metastasis risk reaches 28% [6,7].

The use of computed tomography scan (CT scan) and magnetic resonance imaging (MRI) was proposed by many authors to improve the detection rate of small sizing ccRCC [1,8]. However, these are too expensive to be applied to screen for the population at risk. Other studies have focused on laboratory techniques, with promising results [9].

MicroRNAs (miRNAs) are a class of non-coding single-stranded RNAs of about 22 nucleotides. They regulates gene expression in a sequence-dependent manner, acting via direct interaction with the 3′-untranslated region and leading to transitional suppression or degradation of target mRNA. Moreover, miRNAs control the transcription and splicing of target genes [10,11] and are involved in the regulation of about 60% of all human genes [12]. The location of miRNAs in cancer-related genomic regions or fragile sites suggests their potential role in carcinogenesis and, in the last years, many studies showed miRNAs as potential markers for RCC as well as many other types of cancer [13,14,15,16]. The role of miRNAs in cancer, their relative stability in biological fluids, and their resistance to storage in appropriate conditions make them excellent candidates for the development of minimally invasive biomarkers for cancer diagnosis and prognosis [17].

Recently, we conceived a new diagnostic method for ccRCC consisting of the evaluation of some urinary miRNAs [18]. Briefly, initially, we study six miRNA chosen because potentially overexpressed by ccRCC tumor cells according to bioinformatic analysis and potentially having a high chance of being markers in the urine of ccRCC patients according to an algorithm set up by us. Their expression levels were tested on 14 samples of ccRCC and the correspondent adjacent kidney tissue (non-cancerous) by using the reverse transcriptase quantitative PCR (RT-qPCR). The best results were obtained by the amplification of miR-122-5p, miR-1271-5p, and miR-15b-5p. Therefore, the presence of the chosen miRNAs and controls (miR-16-5p, Cel-miR-39-3p, and miRTC) was evaluated on the urine of 13 patients with ccRCC and 14 healthy subjects (HSs). Interestingly, the mean amount of miR-122-5p was 3.9 log2 higher in the urine of patients with ccRCC as compared to that of HSs. Despite the impressive mean overexpression of miR-122-5p, the resulting area under the Receiver Operating Characteristic (ROC) curve was equal to 0.82 (sensitivity equal to 92% and specificity equal to 64%), suggesting that the urinary miR-122-5p cannot be used alone to test the presence of ccRCC in a subject. Therefore, we try to improve the AUC value by combining the Ct values of miR-122-5p, miR-1271-5p, and miR-15b-5p and that of the internal controls. For each sample, the value of 12 parameters was calculated considering each miRNA alone or normalized with each of the three internal controls. The seven parameters appearing to be the best for discriminating patients and healthy donors were chosen using the black-box testing method and the final value of the sample was equal to the sum of the values of the seven parameters (7p-urinary score). Using the 7p-urinary score, the ROC curve showed an AUC of 0.96 (*p* < 0.0001), a sensitivity of 100%, and a specificity of 86%, suggesting that the combination of the seven parameters improves the predictivity and can be used to test the presence of ccRCC in a subject.

It is well-known that every new method shows higher sensibility and specificity in the discovery cohort than in the real world. Therefore, in this study, we sought to validate the method in an independent validation cohort and confirmed the validity of the previously reported diagnostic algorithm.

## 2. Materials and Methods

### 2.1. Subjects

The Local Ethics Committee approved the research (no. 3193/18), and subsequently, the informed written consent was acquired by all the subjects. The procedures and techniques were carried out according to significant guidelines, and national/international rules and laws. The collection of urinary samples was executed in agreement with laws and under proper ethical conditions. All data and urinary samples from enrolled subjects were evaluated anonymously.

For the independent validation cohort (from October 2018 to January 2021), 28 urinary specimens were acquired from ccRCC patients before nephrectomies, which were performed at the Division of Urologic Clinic, Santa Maria della Misericordia Hospital, University of Perugia, Italy. All diagnoses were histologically confirmed. Total nephrectomy was executed in 7 patients (25%), while partial nephrectomy in 21 patients (75%). The control urine specimens were obtained from 28 healthy subjects (HSs) matched for sex and age.

For both patients with ccRCC and HSs, providing informed consent, the inclusion criterium was age over 30 years and the exclusion criteria were the presence of urinary stones or infections, diabetes, chronic liver or kidney diseases, severe kidney failure (eGFR less 30 mL/min), and active neoplastic form in the past three years. For every patient included in this study, the presence of comorbidities was evaluated using the Charlson Comorbidity Index [19]. In the evaluation of the Charlson Comorbidity Index, the presence of ccRCC was not considered. The baseline renal function was assessed with the evaluation of the serum creatinine and Glomerular Filtration Rate (eGFR). The population characteristics, smoking status, kidney function, and comorbidities of HSs and patients with ccRCC are reported in Table 1.

No differences were observed between HSs and patients with ccRCC concerning demographics, smoking status, renal function, and Charlson Comorbidity Index. The non-significant higher value of the Charlson Comorbidity Index in the patients with ccRCC as compared to HSs is mainly due to a non-significant higher mean age, more frequent cardiovascular diseases (7 vs. 1), gastric ulcer (2 vs. 0), BPCO (2 vs. 0) and past tumors (6 vs. 3) without any sign of tumor mass or laboratory pathological value.

The mean (±S.D.) clinical size of tumors was 4.20 ± 2.42 cm (range: 1.30–10.30 cm). The R.E.N.A.L. nephrometry score was used to stratify patients according to kidney disease [20]. The clinical and pathological data of ccRCC tumors are described in Table 2.

### 2.2. Amplification of MiRNAs from Urine

Urinary samples from all subjects were drawn in the afternoon (2 PM) after hospital admittance. In particular, ccRRC urines were collected one day before the surgical intervention. As reported in the previous study [18], 25 mL of urine specimen was mixed within 4 h with 0.5 mL of urine preservation solution (Norgen Biotek, Thorold, ON, Canada) and stored in the fridge at 4 °C.

A total of 200 μL of whole urine, without any centrifugation, was taken for total RNA extraction with miRNeasy Micro Kit (Qiagen, Hilden, Germany), adding 5.6 × 10^8^ copies of exogenous Cel-miR-39-3p spike-in control (Qiagen). The RNA concentration and purity were estimated by Nanodrop 2000c spectrophotometer (Thermo-Fisher Scientific, Waltham, MA, USA).

The miScript^®^II RT kit (Qiagen) was used for RNA reverse transcription. Subsequently, the miScript^®^PreAMP PCR kit (Qiagen) was employed for the pre-amplification assay, and the obtained pre-amplified cDNA was diluted 20-fold. The expression levels of the tested miRNAs, miR-122-5p, miR-1271-5p, and miR-15b-5p, and a panel of 3 controls, miR-16-5p, Cel-miR-39-3p, and miRTC, were evaluated by RT-qPCR in triplicate by using ABI 7300 cycler (Thermo-Fisher Scientific, Waltham, MA, USA) and miScriptSYBR^®^Green PCR Kit (Qiagen). No template controls were also included.

The reaction conditions were 95 °C for 15 min and 38 cycles at 94 °C for 15 s, 60 °C for 30 s, and 70 °C for 34 s. The amount of miRNA was evaluated considering the relative concentration of target in the reaction (threshold cycle, Ct) (Appendix A) and using the ΔCt method (Ct (miRNA target) − Ct (internal control)) when indicated. The Ct of the samples with melting curves altered was evaluated as not amplified.

### 2.3. Parameters of 7p-Urinary Score

First, the mean, the standard deviation (S.D.), and the ranges (mean ± S.D.) of the urinary Ct and ∆Ct value of each parameter of the 7p-urinary score were calculated.

Appendix A shows that the mean values of parameters #1, #2, and #7 were lower in ccRCC patients than in HSs, while the mean values of parameters #3, #4, #5, and #6 were greater in ccRCC patients than in HSs.

The parameters #1, #2, and #7 with a lower mean value in ccRCC patients than in HSs were subtracted from the mean + S.D. value of Ct or ΔCt values in ccRCC patients. On the contrary, in parameters #3, #4, #5, and #6 showing a higher mean value in patients with ccRCC than in HSs, the mean − S.D. ΔCt value of ccRCC patients was subtracted from the mean of ΔCt values (Appendix A). Then, for each HS and ccRCC patient, the sum of seven parameters was determined as indicated in Appendix A.

### 2.4. Statistical Analysis

Statistical analyses, which included Fisher’s exact test, Kolmogorov–Smirnov (KS) normality test, unpaired *t*-test, Mann-Whitney test, Receiver Operating Characteristic (ROC) curve, and correlation analysis were executed using Prism 8.0.1 (GraphPad Software, San Diego, CA, USA).

After establishing the distribution of the population by the use of the KS test, other statistical processing was carried out. To estimate the differences between the two groups, *p*-values were computed by the unpaired *t*-test when the samples passed the KS for normality or using the Mann-Whitney test when they failed. To correlate two parameters, the correlation coefficient, and *p*-values were computed by the Pearson correlation test when the samples passed the KS for normality or using the Spearman correlation test when they failed.

## 3. Results

### 3.1. Expression of Urinary MiR-122-5p, MiR-1271-5p, and MiR-15b-5p

As a first validation step, we tested the levels of urinary miR-122-5p, miR-1271-5p, and miR-15b-5p and the three internal controls (miR-16-5p, Cel-miR-39-3p, and miRTC) in 28 urinary samples from patients with ccRCC and 28 samples from HSs.

To avoid false positive and false negative results, we used the following procedures: First, miRNA amplification was performed in triplicate using RT-qPCR. The urine sample was considered to be good enough for testing only if the Ct value of Cel-miR-39-3p was less than 35. Since four samples of ccRCC patients and five samples of HSs showed a Ct value > 35 for Cel-miR-39-3p, we evaluated the modulation of the tested miRNAs in 24 samples from patients with ccRCC and 23 samples from HSs. Then, to be certain that a miRNA was not expressed in a urinary sample when evaluating the miRNAs (test and internal controls), a second independent amplification of the miRNAs was performed to check if their Ct values were greater than 38 in the first amplification. If both RT-qPCR amplifications showed a Ct value ≥ 38, the miRNA was considered not present in the urine sample, and its Ct was set to 40 by convention.

Table 3 shows that miR-122-5p, miR-1271-5p, and miR-15b-5p were overexpressed in the urine samples of patients with ccRCC compared to HSs. Specifically, the mean expression of miR-122-5p was 2.55 Log2-fold higher in the urine of patients with ccRCC (*p* = 0.0192), confirming the data obtained in the discovery cohort [18]. The mean expression of miR-1271-5p was 1.18 Log2-fold higher in the urine of ccRCC patients (*p* = 0.0645), and overexpression slightly lower than that observed in the discovery cohort (1.74 Log2-fold).

In the discovery cohort [18], the amount of miR-15b-5p was similar in the urine from the patients and HSs, while in this validation cohort, the amount of miR-15b-5p was 2.91 Log2-fold higher in the urine from the patients with ccRCC than in that from HSs, but the difference was not significant (*p* = 0.0817), suggesting that miR-15b-5p expression had high inter-subject variability.

The differences in expression of the three miRNAs between the discovery cohort and the validation cohort may be due to the differences in several parameters, including patient-dependent variables (e.g., genetic and anatomy), sample-dependent variables (e.g., urine concentration and urine pH), and laboratory-dependent variables.

### 3.2. Predictive Power of MiRNAs Taken Separately

To evaluate whether the three miRNAs allow us to distinguish ccRCC patients from HSs, we calculated the median Ct value of miR-122-5p, miR-1271-5p, and miR-15b-5p considering both ccRCC patients and HSs as the overall population. Figure 1 shows that miR-122-5p, miR-1271-5p, and miR-15b-5p can differentiate HSs from patients with ccRCC. Indeed, 61% of the Ct values of miR-122-5p and miR-1271-5p in ccRCC patients was lower than the median Ct. Similar results were obtained with miR-15b-5p, for which 67% of Ct values was lower in ccRCC patients than the median Ct.

The AUC of miR-122-5p was significant and showed a value of 0.70 (Appendix A), almost comparable to that obtained by us in the discovery cohort (AUC of 0.80). Meanwhile, the AUCs of miR-1271-5p and miR-15b-5p were 0.66 and 0.65, respectively (Appendix A) and the data are not significant (*p* = 0.0641 and *p* = 0.0810).

### 3.3. Predictive Power of the 7p-Urinary Score

In our previous published study, we decided to use more than one parameter because the predicting power based on the expression level of each miRNA considered alone was low [18]. Similarly, in the validation cohort, the only miRNA demonstrating predicting power was miR-122-5p, but the AUC, sensitivity, and specificity were not satisfactory.

The set algorithm considered seven parameters derived from the evaluation of the urinary levels of miR-122-5p, miR-1271-5p, and miR-15b-5p and three internal controls, all reported as Ct or ΔCt (normalized with the specified internal control) values of the miRNA: parameter #1, miR-1271-5p; #2, miR-122-5p/miR-16-5p; #3, miR-122-5p/miRTC; #4, miR-1271-5p/miR-16-5p; #5, miR-1271-5p/miRTC; #6, miR-15b-5p/miRTC; #7, miR-15b-5p/Cel-miR-39-3p. The 7p-urinary score of each urine sample was derived as the sum of the seven values of the parameters. Forecasting inter-laboratory differences, the method elaborates data from the urinary samples of the patients and a laboratory-specific cut-off (see Material and Methods and [18]).

The mean 7p-urinary score was found to be significantly different between ccRCC patients and HSs (*p* < 0.0001), as seen in Figure 2a. In addition, Appendix A shows that the mean 7p-urinary score in ccRCC patients with grade one tumor was similar to that of ccRCC patients with higher grades of malignancy (grade two and grade three). Moreover, the 7p-urinary score was similar in patients with tumors of small size (<5 cm diameter) and large size (>5 cm diameter) (Appendix A).

Figure 2b shows that in the validation cohort, the published method differentiates HSs from patients with ccRCC except for one false negative and eight false-positive results. Figure 3 shows that the ROC curve of the 7p-urinary score has an AUC of 0.81 and *p* = 0.0003. Using the cut-off of −14.76, the 7p-urinary score shows a sensitivity of 96% and specificity of 65% in the validation cohort.

### 3.4. Strength of 7p-Urinary Score

A screening method must be appliable to real-world subjects and patients that are very heterogeneous as far as concerns sex, age, and lifestyle. Therefore, we tested if the 7p-urinary score change according to the population tested. Figure 4 show that the 7p-urinary score is similar in HSs that are male or females (left panel a), adult or elderly (left panel b), and nonsmoker or smokers/ex-smokers (left panel c). The same figure shows no differences within patients with ccRCC (right panels a, b and c).

We hypothesized that patients affected by urinary stones, urinary infection, or kidney diseases may give biased results and these patients were not included in the study. We also excluded patients with diabetes, chronic liver disease, and other neoplastic forms in the past 3 years. However, we included patients with several co-morbidities, such as cardiovascular (including past IMA, heart failure, peripheral vascular disease, and hypertension), pulmonary (including BPCO and asthma), and gastrointestinal diseases. Moreover, we included patients with low eGFR (not less than 30 mL/min). Even if co-morbidity index and eGFR were not significantly different between HSs and patients with ccRCC (Table 1), we analyzed whether the co-morbidity index correlated with the 7p-urinary score to test whether the 7p-urinary score is dependent, at least in part, on these parameters. Figure 5 shows that the 7p-urinary score correlates neither with the comorbidity index nor eGFR in healthy subjects and patients with ccRCC.

Therefore, the 7p-urinary score appears to be independent of sex, age, lifestyle, and comorbidities, except the comorbidities that we did not test (see above and Methods section).

### 3.5. Unsuccessful Attempts to Improve the Predictive Power of the 7p-Urinary Score

Due to overfitting, we expected to find lower AUC, sensitivity, and specificity of the method in the validation cohort than in the discovery cohort and indeed we found it. However, since the seven parameters were chosen from among the 12 parameters empirically (black-box testing method), to obtain the best possible results, we re-evaluated each of the seven parameters to evaluate whether any of them were chosen in error. In this case, we would have verified whether by removing a given parameter the new “urinary score” was more predictive than the previous one in both the discovery and validation cohorts. In particular, we have focused our attention on parameters #4 and #7, because they showed quite different results in the discovery and validation cohorts (Appendix A). In particular, the mean value of parameter #4 was lower in ccRCC patients than in HSs in the discovery cohort and higher in the validation cohort. In addition, the mean value of parameter #7 in ccRCC patients in the discovery cohort was markedly different from that in the validation cohort. However, when considering only parameters #1, #2, #3, #5, and #6 to calculate the score, the predicting power lowered compared to that obtained when including the seven parameters, with an AUC of 0.74, a sensitivity of 92%, and a specificity of 57%. A similar result was obtained considering all seven parameters except one. Therefore, we concluded that in the validation cohort also, the seven parameters contribute to improving the predictive value of the algorithm and all seven parameters must be considered in the scoring.

We also tested whether the AUC, sensitivity, and specificity were improved by separately considering males and females. When considering only females, we obtained an AUC of 0.78, a sensitivity of 89%, and a specificity of 73%. Meanwhile, when considering only males, we obtained an AUC of 0.72, a sensitivity of 87%, and a specificity of 75%. Therefore, the AUC, sensitivity, and specificity of the 7p-urinary score were not improved by separately considering females and males.

In conclusion, the 7p-urinary score seems to offer the best possible result using the miR-122-5p, miR-1271-5p, miR-15b-5p, and the three internal controls. Indeed, none of the evaluated changes resulted in any improvements in the AUC, sensitivity, or specificity. Even the selection of a subgroup of subjects did not improve the results.

## 4. Discussion

In a previous study, we investigated the possibility of performing ccRCC diagnosis by evaluating the presence of urinary miR-122-5p, miR-1271-5p, and miR-15b-5p and a set algorithm called the 7p-urinary score, which considers seven parameters derived from the amplification of the three investigated miRNAs and internal controls. In the present study, using an independent cohort, we confirmed what we found in the previous study, even if AUC value (0.81 vs. 0.96), sensitivity (96% vs. 100%), and specificity (65% vs. 86%) were lower.

The lower predictive values found in the independent cohort is due to overfitting introduced in the previous study [18] (always observed when a new predictive method is described in a discovery cohort) and the inter-subject variability of each parameter, suggested by the high standard deviation of values, particularly in HSs (Appendix A). Variability is observed with each miRNA. The levels of urinary miR-122-5p in the present cohort differentiate patients with ccRCC from HSs with an AUC of 0.70, a sensitivity of 83%, and a specificity of 48% (Appendix A); however, values are lower than those found in the previous study (e.g., AUC of 0.82) [18]. The same happened with miR-1271-5p. On the contrary, the AUC of urinary miR-15b-5p was higher in this study (0.65) than in the previous (0.59) [18]. Parameter instability appears to be intrinsic to miRNAs (at least the ones we used) and is due to high intersubjective variability and, likely, non-optimal laboratory reproducibility. Interestingly, using multiple parameters helps reduce variability and improves sensitivity and specificity in the ccRCC diagnosis. Indeed, the AUC of the best parameter considered alone (miR-122-5p) ranges between 0.70 (present study) and 0.82 (previous study) [18], and the AUC of the 7p-urinary score ranges between 0.81 (present study) and 0.96 (previous study) [18].

Several studies have evaluated the potential diagnostic role of miRNAs in RCC. Von Brandenstein et al. [21] showed a higher amount of miR-15a in the urine of RCC patients. Mytsyk et al. [22] compared the expression of urinary miR-15a in RCC patients, patients with benign renal tumors, and HSs, obtaining a specificity and sensitivity equal to 98% and 100%, respectively. Petrozza et al. [23,24] and Li et al. [25] found that miR-210-3p was upregulated in the urine of ccRCC patients. In particular, urinary miR-210-3p showed an AUC of 0.76 with a sensitivity of 58% and specificity of 80% [25]. Moreover, Fedorko et al. [26] demonstrated that the evaluation of urinary miR-let-7a allows for differentiating RCC patients from controls with a sensitivity of 71% and specificity of 81%.

To ensure that miRNA quantification is not affected by technical variability, in this study, we considered the urine samples reliable only when the Ct values of Cel-miR-39-3p were less than 35. Spike-in Cel-miR-39-3p is an optimal candidate for the quality check of samples because it can monitor both RNA purification and reverse transcription efficiencies. Petrozza et al. [23,24] and Li et al. [25] used the same internal control. In the present study, we evaluated the levels of two additional internal controls that were used to calculate the 7p-urinary score: miR-16-5p as endogenous internal control and miRTC, which was added during the reverse transcription reaction. Based on our results, we conclude that using three internal controls allowed us to normalize both the quantity and quality of the extracted RNA and the quality of the reactions (reverse transcription, preamplification, and amplification), ensuring a certain reproducibility of the data. Other internal controls appear to be less appropriate. For example, Mytsyk et al. [22] used the small nuclear RNA U6 (RNU6) to normalize data obtained from urine samples. RNU6 is the most commonly used endogenous control for the analysis of miRNA expression in cells and tissues [27,28]. However, RNU6 is a small nucleolar RNA, and its release into biological fluids appears unlikely; for this reason, the use of RNU6 for the normalization of data obtained from urine may be inappropriate [18].

To date, our method is one of the few that has been validated using an independent validation cohort. In fact, among the studies on the use of urinary miRNAs for the diagnosis of RCC, only Petrozza et al. [23,24] have validated their results using a second independent cohort. Another strength of our study is the fact that the validation cohorts include a heterogeneous group of healthy subjects and patients, and statistic demonstrates that the 7p-urinary score is not different in subjects of different sex, age, renal function, and comorbidities.

Our method could be used to screen subjects at risk for ccRCC, and when positive, subjects could be further evaluated by imaging techniques to confirm or exclude the presence of ccRCC tumors. However, the relatively low specificity of our method combined with the relatively low incidence of ccRCC in the population could result in a high number of false-positive subjects, substantially increasing the cost of the screening procedure, even if performed in a population at high risk of ccRCC. However, our study is the proof of concept that combining more parameters increases sensitivity and specificity of diagnosis and encourages the study of more parameters that may be useful to improve sensitivity and specificity.

There are six limitations of our study. First, the validation cohort is relatively small. We do not know if the prediction of our method would be decreased or increased using a larger cohort. However, reaching significant values with the relatively small cohort we studied suggests that the method is reliable. Second, subjects with urinary stones, infections, diabetes, liver and kidney chronic diseases, and other recent/present neoplastic forms were not included in the study. Therefore, people with the abovementioned comorbidities cannot be tested in an eventual screening of the population. Third, in our study, 16% of the screened subjects were not included in the final evaluation because the quality of the RT-qPCR amplification was not good enough. Therefore, it is possible that approximately one-fifth of the subjects evaluated using our method may not be able to obtain a result in the real world. Fourth, the comorbidity index is higher (even if the difference is not significant) in ccRCC patients than in healthy subjects (3.32 vs. 2.00) and the difference of 7p-urinary score may be due, at least in part, to other diseases. However, we did not find a correlation between comorbidities and 7p-urinary score, possibly suggesting that comorbidities do not influence the 7p-urinary score. Another limitation of the method is that it is unclear whether patients with early-stage tumors will be positive in our test. In the previous study [18], we found that the 7p-urinary score demonstrated higher predictive power in patients with larger tumors than smaller tumors, suggesting that the signals in small tumors are at low levels. On the contrary, we found here that the 7p-urinary score value and its predictive power are independent of the tumor mass. Finally, the best way to validate a method is to promote its testing by another group of the author. On the contrary, the authors of this manuscript are almost the same as those of the previous one.

## 5. Conclusions

The 7p-urinary score is a potentially useful tool for early ccRCC diagnosis using a noninvasive technique. However, it must be improved by increasing the number of parameters.

## Figures and Tables

**Figure 1 cancers-14-01112-f001:**
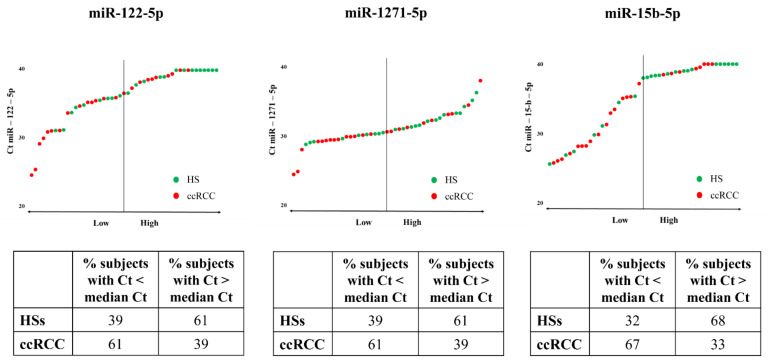
Expression levels of miR-122-5p, miR-1271-5p, and miR-15b-5p in the urine of HSs and ccRCC patients. Ct values of miR-122-5p, miR-1271-5p, and miR-15b-5p in the HS urine (green dots) and ccRCC patients (red dots). The tables indicate the percentage of HSs and patients with ccRCC showing Ct values lower and higher than the median Ct. The median Ct values were 36.62, 30.70, and 37.96 for miR-122-5p, miR-1271-5p, and miR-15b-5p, respectively.

**Figure 2 cancers-14-01112-f002:**
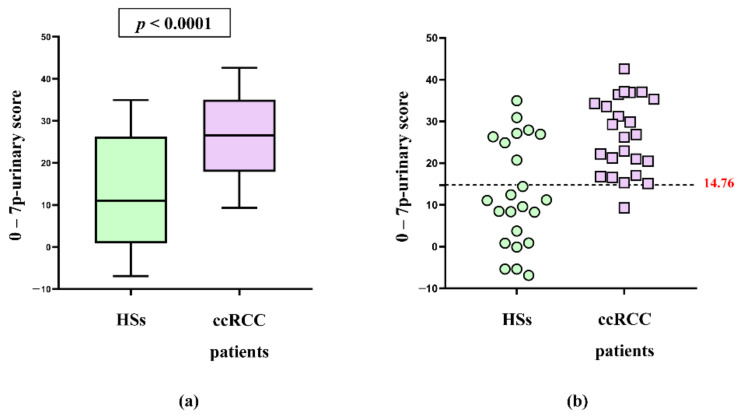
The 7p-urinary scores of patients with ccRCC and HSs. (**a**) Box and whiskers plots of 7p-urinary score of HSs and patients with ccRCC are shown. The mean 7p-urinary score of HSs is significantly different to that of patients with ccRCC (unpaired *t*-test). (**b**) In the validation cohort, the best cut-off for differentiating patients with ccRCC and HSs resulted to be −14.76 (dotted line). In the panel, the samples with a 0-(7p-urinary score) value < 14.76 should belong to HSs, and the samples with a 0-(7p-urinary score) value ≥ 14.76 should belong to patients with ccRCC.

**Figure 3 cancers-14-01112-f003:**
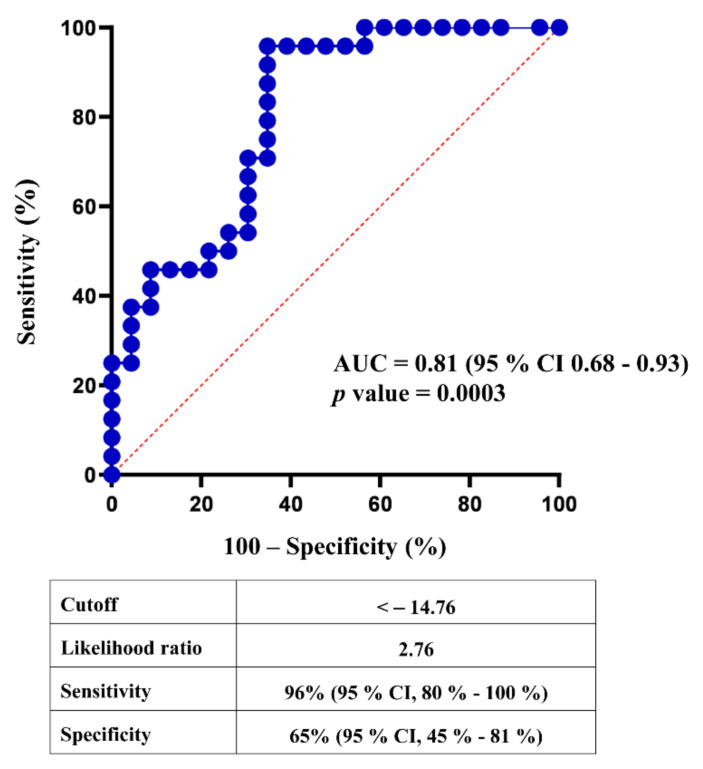
Predictive power of 7p-urinary score. ROC curve was calculated with the sum of parameters #1, #2, #3, #4, #5, #6, and #7 as shown in Appendix A.

**Figure 4 cancers-14-01112-f004:**
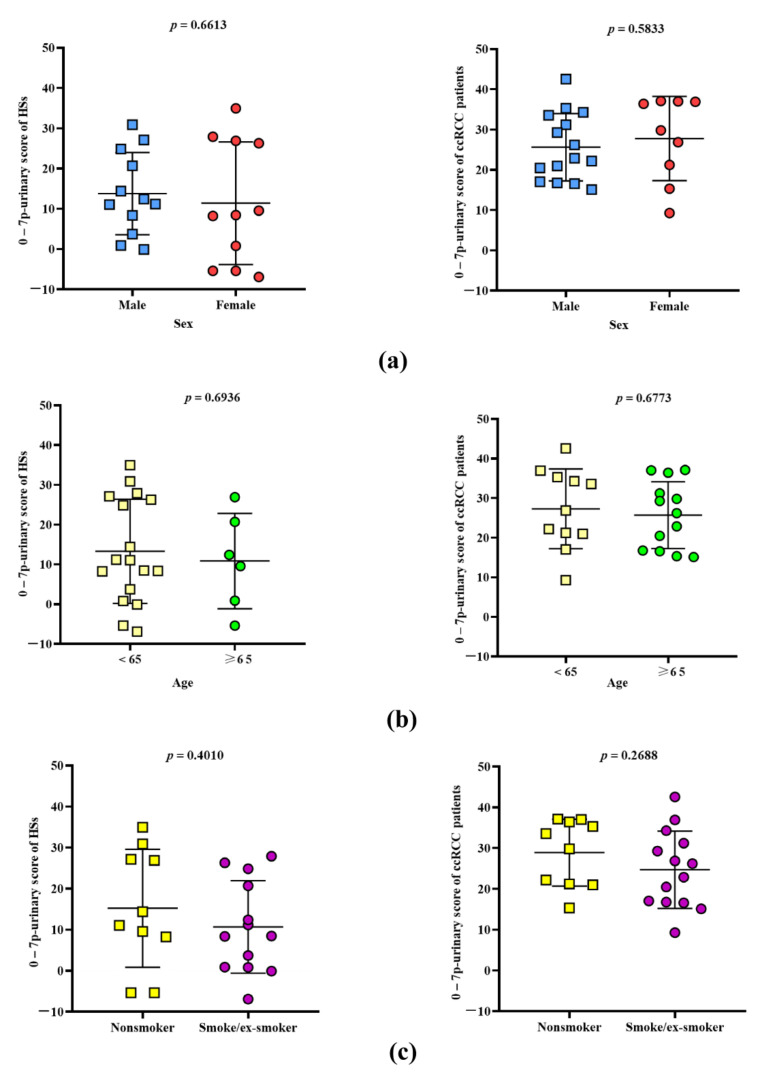
The 7p-urinary score of HSs and ccRCC patients with different demographic characteristics and lifestyles. 7p-urinary score values in HSs (**left**) and patients with ccRCC (**right**) male and females (**a**), with age <65 years and ≥65 years (**b**), and no smoker and smoker/ex-smokes (**c**). Mean values ±1 SD is shown. Differences between the groups were evaluated using the unpaired *t*-test.

**Figure 5 cancers-14-01112-f005:**
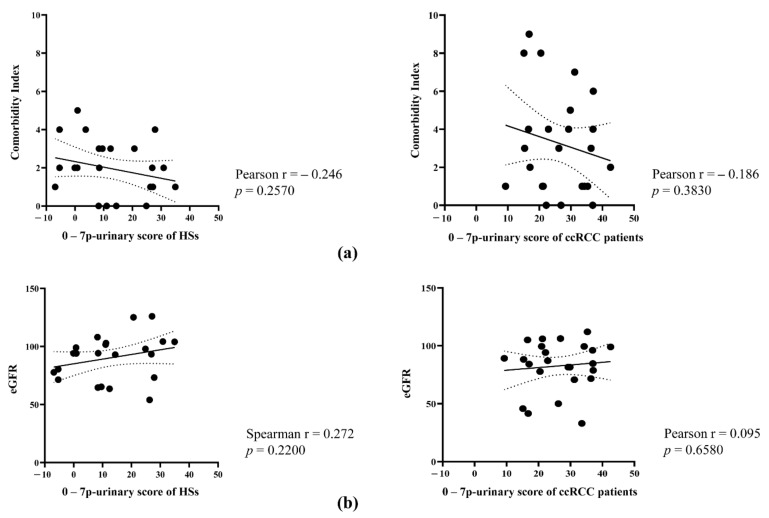
Correlation of Charlson Comorbidity Index and eGFR with the 7p-urinary score. (**a**) Correlation of 7p-urinary score with Charlson Comorbidity Index in HSs (**left**) and patients with ccRCC (**right**). Pearson correlation coefficient and *p*-value are reported. (**b**) Correlation of 7p-urinary score with eGFR both in HSs (**left**) and patients with ccRCC (**right**). Pearson correlation coefficient (if KS test was passed) or Spearman correlation coefficient (if KS test failed) and *p*-value are reported.

**Table 1 cancers-14-01112-t001:** Demographic characteristics, smoking status, renal function, and impact of comorbidities of healthy subjects (HSs) and patients with ccRCC whose urine was studied. Appropriate statistical tests were applied to evaluate the difference between HSs and ccRCC patients.

	HSs (*n* = 28)	ccRCC Patients (*n* = 28)	Passed KS ^1^ Test	Statistics
**Sex**				
Male	15	17	Not Applicable	Fisher’s exact test *p* = 0.7875
Female	13	11		
**Age**				
Mean ± S.D. ^2^	60.79 ± 11.73	65.00 ± 12.96	Yes	Unpaired *t*-test *p* = 0.2075
Range	39–88	41–84		
**Smoke**				
Smoker/ex-smoker	18	16	Not Applicable	Fisher’s exact test *p* = 0.7848
Nonsmoker	10	12		
**Serum creatinine (mL/dL)**				
Mean ± S.D. ^2^	0.80 ± 0.18	0.94 ± 0.34	No	Mann-Whitney test *p* = 0.1504
Range	0.50–1.21	0.63–2.16		
**eGFR ^3^ (EPI-CKD ^4^ equation, mL/min × 1.73 m^2^)**				
Mean ± S.D. ^2^	90.59 ± 18.67	81.65 ± 20.19	Yes	Unpaired *t*-test *p* = 0.0979
Range	54–126	33–112		
**Charlson Comorbidity Index**				
Mean ± S.D. ^2^	2.00 ± 1.41	3.32 ± 2.58	No	Mann-Whitney test *p* = 0.0719
Range	0–5	0–9		

^1^ Kolmogorov-Smirnov. ^2^ Standard Deviation. ^3^ Glomerular Filtration Rate. ^4^ Chronic Kidney Disease Epidemiology Collaboration.

**Table 2 cancers-14-01112-t002:** Clinicopathological data of ccRCC tumors of the cohort.

Fuhrman Grade		
	G1	4
	G2	23
	G3	1
	G4	0
**T stage**		
	T1	24
	T2	2
	T3	1
	T4	0
	Unknown	1
**M stage**		
	M0	26
	M1	2
**N stage**		
	N0	27
	N1	1
**Tumor mass size ^1^ (cm)**		
	Mean ± S.D. ^2^	4.20 ± 2.42
	Range	1.30–10.30
**R.E.N.A.L. score**		
	Mean	7.28 ± 1.97
	Range	4–11
**Type of nephrectomy**		
	Partial (%)	21 (75.00%)
	Total (%)	7 (25.00%)

^1^ Largest diameter evaluated by CT scan. ^2^ Standard Deviation.

**Table 3 cancers-14-01112-t003:** Mean modulation levels of miR-122-5p, miR-1271-5p, and miR-15b-5p in the urine samples of ccRCC patients compared to healthy subjects (HSs).

miRNA	Fold-Change ^1^ (Log_2_)	Passed KS Test ^2^	*p*-Value (Unpaired *t*-test ^2^)
miR-122-5p	2.55	No	0.0192
miR-1271-5p	1.18	No	0.0645
miR-15b-5p	2.91	No	0.0817

^1^ Overexpression of miRNA was calculated using the following formula: Mean miRNA Ct (HSs)—mean miRNA Ct (ccRCC). ^2^ Since the samples failed the Kolmogorov-Smirnov (KS) test for normality, the Mann-Whitney test was used.

## Data Availability

Raw data supporting reported results can be found in the Appendix A.

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
