# Peer review of "Validation in an Independent Cohort of MiR-122, MiR-1271, and MiR-15b as Urinary Biomarkers for the Potential Early Diagnosis of Clear Cell Renal Cell Carcinoma"

_cancers, 2022, doi:10.3390/cancers14051112_

Round 1

Reviewer 1 Report

In this manuscript, the authors tested the hypothesis that certain small RNAs could diagnose ccRCC in human patients. Using a patient’s urine is a noninvasive method and therefore is always welcomed for clinical tests.

The authors observed a sensitivity of 96% and a specificity of 65% using their in-house algorithm, named 7p-urinary score, that utilized urine miRNA, for the detection of ccRCC.

The authors gave a very good description of the limitations of the study in the discussion.

My one weakness of the study was that there was not a description of how the 7p-urinary score algorithm was developed and the hypothesis behind it (even though I understand there was a previous manuscript). While reading this manuscript the use of the sum of the seven different miRNA values seemed random. I think a better description of why this was utilized would help the readers and better put value on the algorithm.

Can the authors explain or postulate why parameters #4 and #7 showed different results between the two cohorts? Also whether the authors are thinking about reevaluating the algorithm to reconsider more stable parameters for a better fit?

Reviewer 2 Report

This is potentially interesting validation study. However validation cohort is rather small and optimally, validation should be done by different researchers among different population subjects.    Main problem of this study is:  - Comorbidity Index is much higher in ccRCC patients than in healthy subjects (4,71 vs. 1,86). Thus, whole difference in urine miRNA could be due to other pathology (e.g. hypertension, smoking...) and not related to ccRCC. It is not clear (rows 89-91) if individuals with stones, infections, diabetes, kidney or liver disease were excluded from HS and ccRCC group? If yes, what other comorbidities had ccRCC patients?  Authors need to address this issue. This also needs to be discussed as a limitation of the analysis.    - Why only 23 healthy subjects and 24 ccRCC patients (out of 28) were included in Figure 1 and Table S1, S2 analysis? This needs to be clarified since 5 and 4 individuals could change whole statistics.  - It needs to be discussed why those miRNAs and those particular internal controls were selected. - It needs to be stated how delta Ct value was calculated. Is it ratio of respective Ct values? - Why miR-15 in #7 was compared with cel-miR-39 and not with miR-16 like #2 and #4 (and vice versa, why #2 and #4 were not tested with cel-miE-39) in TS1 and 2? - Mean value for HS in Figure 2a needs to be recalculated since my calculation comes with mean of 12,6 and according to figure it is close to 10 (mean for ccRCC is 26,4, which fits to figure).   - Discussion is rather lengthy and last sentence in Conclusions is unnecessary.  - Font size in rows 173-176 need to be adjusted.

Reviewer 3 Report

In the paper authors analyzed potential utility of 3 miRNAs for the detection of clear cell renal cell carcinoma. Idea of the study is quite promising, however, paper suffers from limiatations:

  1. First, authors tested small and heterogenous group of patients, what make difficult statistical interpretation and clinical application. Second, the proposed algorithm is a bit confusing and using only signature of 3 miRNAs it is possible to create more convenient form.
  2.  Please check by appropriate tests differences between study and control grup.
  3. Do expression of miRNAs correlated with clinical-demographic data of patients?
  4. There should be more careful explanation why did authors chose those miRNAs for analysis.

Round 2

Reviewer 2 Report

None

Reviewer 3 Report

Authors responded to addressed comments and did effort to improve paper quality.
